# MicroRNAs in Ocular Infection

**DOI:** 10.3390/microorganisms7090359

**Published:** 2019-09-17

**Authors:** Shunbin Xu, Linda D. Hazlett

**Affiliations:** Department of Ophthalmology, Visual and Anatomical Sciences, Wayne State University, School of Medicine, Detroit, MI 48201, USA

**Keywords:** microRNAs (miRNAs), ocular infection, trachoma, river blindness, fungal keratitis, bacterial keratitis, *pseudomonas aeruginosa* (PA), herpes simplex stromal keratitis (HSK)

## Abstract

MicroRNAs (miRNAs) are small, non-coding, regulatory RNA molecules and constitute a newly recognized, important layer of gene-expression regulation at post-transcriptional levels. miRNAs quantitatively fine tune the expression of their downstream genes in a cell type- and developmental stage-specific fashion. miRNAs have been proven to play important roles in the normal development and function as well as in the pathogenesis of diseases in all tissues and organ systems. miRNAs have emerged as new therapeutic targets and biomarkers for treatment and diagnosis of various diseases. Although miRNA research in ocular infection remains in its early stages, a handful of pioneering studies have provided insight into the roles of miRNAs in the pathogenesis of parasitic, fungal, bacterial, and viral ocular infections. Here, we review the current status of research in miRNAs in several major ocular infectious diseases. We predict that the field of miRNAs in ocular infection will greatly expand with the discovery of novel miRNA-involved molecular mechanisms that will inform development of new therapies and identify novel diagnostic biomarkers.

## 1. Introduction

MicroRNAs (miRNAs) are small, non-coding regulatory RNAs, about 20–24 nucleotides in size [1,2,3,4]. miRNAs were first discovered in 1993 in *Caenorhabditis elegans* [2,3]. Increasing numbers of miRNAs have been identified in all studied metazoans. On the basis of miRBase Release 22.1 (October, 2018; www.mirbase.org), there are at least 2654 mature miRNAs (1917 precursors) in humans and 1978 mature miRNAs (from 1234 precursors) in mice [5].

miRNAs constitute a relatively recent layer of gene-expression regulation at a post-transcriptional level [1,6]. Like most protein-coding genes, miRNA genes reside in the chromosomes and are mostly transcribed by RNA polymerase II as primary transcripts (pri-miRNAs) that are capped, polyadenylated, and spliced, as seen in Figure 1 [7]. As seen in Figure 1A, pri-miRNAs fold into hairpin structures that are cleaved by an RNase III endonuclease, the Drosha/DGCR8 complex, to form 60–70 nt stem loop intermediates, known as pre-miRNA, with a 2 nt 3′ overhang, seen in Figure 1C [1,8,9]. Pre-miRNAs are transported from the nucleus to the cytosol by an Exportin 5-dependent mechanism, as seen in Figure 1D. More than one-fourth of the conserved and more than half of the poorly conserved miRNAs reside in the introns of protein-coding genes and are processed to pre-miRNAs through a splicing mechanism with or without involvement of Drosha in the nucleus, as seen in Figure 1B [10,11,12,13]. Pre-miRNAs are cleaved in the cytoplasm by a second RNase III endonuclease, Dicer, to produce double-stranded miRNA:miRNA* duplexes, as seen in Figure 1E,F, that are loaded into the RNA-induced silencing complex (RISC), seen in Figure 1G, where the miRNA* strand is degraded, seen in Figure 1H [14,15]. Mature, single-stranded miRNAs engage in base pairing with target sites, which are typically located in the 3′ untranslated region (UTR) of the transcripts of their downstream genes [1,16]. The seed sequence, nucleotide (nt) 2–8 from the 5′ end of the mature miRNA, is a major determinant of its targets, although other features also influence the targeting specificity [17,18,19,20,21]. miRNA targeting with nearly perfect complementary leads to cleavage of the corresponding mRNAs, as seen in Figure 1I [6,9,22], while binding with imperfect complementary usually results in an translation inhibition of the targeted mRNA, seen in Figure 1J [23]. Mature miRNAs can be degraded by the highly regulated Target RNA-directed miRNA Degradation (TDMD), as seen in Figure 1K [24,25,26,27].

miRNAs account for approximately 1–5% of animal genes [1]. It is estimated that more than one-third of the protein-coding genes in the human genome are subjected to miRNA regulation [28,29]. One miRNA can target and regulate hundreds of downstream genes [20,22,30], while one mRNA can be targeted by multiple miRNAs, forming intricate, fine-tuned gene-expression regulatory networks [31,32]. 

Similar to protein-coding genes, miRNA expression is highly controlled with unique cell type-specificity and temporal patterns [33,34,35,36]. Because different cell types have a different transcriptome at different developmental stages, one miRNA can have different downstream target genes and regulate different signaling pathways in different cell types and at different developmental stages in the same cell type [37,38,39]. Therefore, to define the function of a miRNA must be in the context of specific cell type and developmental stage. 

miRNAs have been proven to play important roles in normal development and function, as well as in the pathogenesis of diseases [40,41,42]. Point mutations in the seed sequences of miRNAs can cause inherited diseases in humans [43,44,45,46,47] and animals [48]. Polymorphisms in pre-miRNAs affecting miRNA biogenesis and in miRNA target sites in the transcripts of their target genes can have a significant functional impact on various biological systems and cause or increase susceptibility to various diseases [49,50,51,52]. miRNAs are quantitative regulators of gene expression; one miRNA often simultaneously targets multiple protein-coding genes in the same signaling pathway or several independent pathways in the same functional network. Although the regulation on each target may be modest, defects in a miRNA can result in simultaneous dysregulation of multiple genes, leading to significant functional consequences when the composite impact passes a threshold [6,22,46,53,54,55]. Therefore, many miRNAs have been identified as viable therapeutic targets for treatment of diseases in human and animals once their role in the pathogenesis of a disease is identified [56]. For example, we and others identified that miR-146 negatively regulates NF-kB activation and downstream inflammation by targeting multiple adaptor proteins of various NF-kB activation pathways [57,58,59]. Because diabetes-induced NF-kB activation and subsequent inflammation contribute to retinal endothelial cell (RECs) death and diabetic retinopathy, we hypothesized that miR-146 is a therapeutic target for treatment of diabetic retinopathy [58,59]. To test this hypothesis, we over-expressed miR-146a in the eyes by intraocular injection of lentivirus expressing miR-146a in a diabetic rat model [60]. We showed that intraocular over-expression of miR-146a resulted in decreased NF-kB activation and inflammation and diabetes-induced retinal functional defects [60]. Various approaches have been developed to enhance or knock down the functions of miRNAs in vivo [61,62,63]. Synthetic oligo-ribonucleotides (ORNs) that mimic the native miRNA duplex are used to enhance the function of a miRNA [64]. To inhibit the function of a miRNA, single-stranded antisense ORNs are used to sequester the endogenous miRNA of interest. A variety of chemical modifications of the ORNs, e.g., 2′-*O*-methoxyethyl (2′-MOE) and locked nucleic acid (LNA) bases, have been developed to enhance the RNA stability for in vivo applications [61,65]. Several clinical trials targeting miRNAs (e.g., miR-34 and miR-16) are ongoing to test their safety in cancer treatment [56]. Recently, miRNAs have been detected in exosomes as a novel mechanism of cell–cell communication and genetic exchange, regulating receipt cell functions [66,67,68,69] as well as providing a new way of miRNA delivery for therapy [70,71,72].

In addition to being novel therapeutic markers, miRNAs in extracellular fluid and exosomes have also emerged as promising diagnostic biomarkers for numerous diseases [73,74,75,76]. In comparison with other types of biomarkers, e.g., mRNAs and protein biomarkers, miRNAs have many unique advantages: miRNAs are remarkably stable in long-term stored biological samples [76,77]. They are smaller in size and are relatively less complex to analyze because they do not go through further modifications like posttranslational modifications in proteins. miRNA transcriptomes can be readily screened by miRNA microarray analysis in hours, with as little as nanograms of input total RNA. Therefore, miRNAs can serve as fast, reliable and sensitive biomarkers for diagnosis and prognosis. For examples, tumor-derived miRNAs in the plasma have been shown to be excellent biomarkers for detection and diagnosis of common human cancers, e.g., lung cancer, breast cancer, hepatocellular cancer, pancreatic cancer, leukemia, colorectal, and prostate cancers [75,76,78]. Circulating miRNAs have been also tested as biomarkers for wide range of other diseases, e.g., diabetes [79], cardiovascular diseases [80,81], neurodegenerative diseases [82], as well as sepsis [83] and other infectious diseases [84].

### 1.1. miRNAs in Ocular Infection

Ocular infectious diseases remain an important cause of blindness worldwide and represent a challenging public health concern [85]. Antibiotics, anti-fungal, and anti-viral drugs remain the mainstay of microbial infections treatment. However, the emergence of strains resistant to antibiotics and other anti-microbial reagents imposes a serious threat to efficient management of ocular infectious diseases. Alternative approach to the treatment of ocular infection is urgently needed. Ocular infection, e.g., microbial keratitis, results in vision loss, secondary to corneal scarring or surface irregularity [85]. Early, accurate diagnosis is the key to efficient treatment of ocular infectious diseases. miRNAs have provided a promising new class of molecules for the development of novel therapeutic targets and diagnostic biomarkers. However, when compared with research in many other systems, miRNA research in ocular infection remains in its early stages. A handful of pioneering studies have explored the roles of miRNAs in the pathogenesis of various ocular infectious diseases and their potential as novel therapeutic targets as well as diagnostic biomarkers. This review will summarize what we have learned from these studies. It will also identify knowledge gaps in the field of miRNAs in ocular infection. We hope this review will provide reference knowledge and guidance for investigators new to this field.

### 1.2. miRNAs in Trachoma

Trachoma is the leading preventable infectious blinding eye disease worldwide (https://www.who.int/news-room/fact-sheets/detail/trachoma). It affects ~150 million people living in the world’s poorest, rural communities and causes an estimated loss of 2.9 billion dollars in productivity annually [86]. It is initiated by infection of the conjunctival epithelium with the obligate intracellular, Gram-negative bacterium, chlamydia trachomatis (Ct). The World Health Organization (WHO) recognize five stages of this disease: (1) trachomatous follicular inflammation (TF) with the presence of five or more follicles in the upper tarsal conjunctiva; (2) trachomatous intense inflammation (TI) with pronounced inflammatory thickening of the tarsal conjunctiva that obscures more than half of the normal deep tarsal vessels; (3) trachomatous scarring (TS) with the presence of scarring in the tarsal conjunctiva causing in-turning of the upper eyelid margin—entropion; (4) trachomatous trichiasis (TT) with at least one eyelash rubbing on the eyeball or evidence of recent removal of in-turned eyelashes; and (5) blinding corneal opacity (CO) [87,88,89]. Although chlamydia infection can be effectively treated by antibiotics, e.g., azithromycin; Mass Drug Administration (MDA) with azithromycin to entire trachoma-endemic districts now is part of the WHO’s public health strategy for trachoma elimination. Repeated infection of the conjunctiva during childhood causes chronic inflammatory response, which can continue without active Ct infection, leading to progressive fibrosis and scarring, and ultimately trichiasis and blindness [89,90]. Therefore, other treatments to control the progression of chronic inflammation is required for efficient management of disease.

Recent reports suggest that host miRNAs are involved in the pathogenesis and progression of trachoma [88,89]. miRNA expression profiling in conjunctival swabs of patients with follicular trachoma, an early stage of the disease, identified at least nine miRNAs (miR-155, -150, -142, -181a/b, -342, -132, -4728, and miR-184) that are differentially expressed between TF (with or without detectable Ct) and normal controls, as seen in Table 1 [88]. Among these, miR-155 and miR-184 demonstrated a direct relationship with the degree of clinical inflammation: miR-155 was upregulated, while miR-184 was downregulated as the severity of clinical inflammation increased [88]. miR-155 has been shown to have wide-range effects on the development and function of immune cells [91]. miR-150 and miR-142 are considered specific to hematopoietic cells [92]. miR-181b and miR-132 are reported to negatively regulate inflammation following toll-like receptor (TLR) or NF-κB activation [88]. miR-342 is involved in inflammatory response, cell proliferation, and cancer [93,94]. miR-4728 is reported to regulate focal adhesion and wound healing [95]. miR-184 is highly enriched in corneal epithelium and plays important roles in corneal development and function [96]. Point mutations in the seed sequence of miR-184 result in syndromes with severe keratoconus [44,45,97]. miR-184 was also reported to be downregulated during an acute corneal injury and restored during wound healing [96]; while in the retina, miR-184 has been shown to be involved in ischemia-induced neovascularization by negatively regulating the Wnt pathway by targeting a Wnt receptor, frizzled-7 [98]. These findings may reflect the host immune response to Ct infection and the wound healing process in the early stages of trachoma [88].

During the TS stage, miRNA profiling identified 82 miRNAs that were differentially expressed in tissue from the conjunctival swabs between healthy and diseased subjects. Among these, miR-147 and miR-1285 were significantly upregulated [89]. Functional annotation of predicted target genes of the differentially expressed miRNAs in trachomatous scarring versus normal conjunctival tissues showed enrichment in pathways involved in fibrosis and epithelial cell differentiation [89].

These reports suggest that miRNAs involved in inflammation and wound healing processes are involved in the pathogenesis of trachoma. Further studies are warranted to identify the molecular mechanisms on how these miRNAs influence the chronic inflammation, fibrosis, and scarring. Once the mechanism is uncovered, a miRNA-based therapeutic strategy may be derived by controlling the inflammation and scarring while enhancing the healing process.

### 1.3. miRNAs in River Blindness

River blindness is a tropical blinding disease caused by infection of the filarial nematode *Onchocerca volvulus* (*O. volvulus*) and is spread by blood-feeding arthropods. It is the second-most common cause of visual impairment and blindness due to ocular infection, after trachoma (*Parasites*. CDC. 21 May 2013) [99]. In Africa, at least 120 million people are at risk of infection [99,100,101]. Approximately 17 million people have been infected with *O. volvulus*, predominantly in Africa, with 1.2 million people suffering from vision impairment or blindness because of *O. volvulus* infection [102]. The infection can affect any part of the eye, from conjunctiva and cornea anteriorly to the uvea and posterior segment, including the retina and optic nerve. In the posterior segment, there is atrophy of the retinal pigmented epithelia and subretinal fibrosis. Autoimmune response is involved. In the anterior segment, the larvae in the cornea and the anterior chamber can be detected by slit lamp examination [103]. The larvae can migrate through the human body without provoking immune response. It is the host’s immune/inflammatory reactions to mostly the dead or dying larvae of *O. volvulus* that cause most of the *Onchocerca*-related pathology, including onchocercal keratitis [103,104]. Anti-helminthic drug, Mectizan™ (Ivermectin) has been used in MDA programs to eliminate onchocerciasis. Although ivermectin treatment has reduced onchocerciasis [105], reliance on a single drug increases the potential for the emergence of ivermectin-resistant *O. volvulus* [106].

Traditionally, microscopic examination of skin biopsies (snips) have been the standard for diagnosis and surveillance of *O. volvulus* infection [107]. Recently, PCR-based assays to detect *O. volvulus* DNA in the skin snips have significantly increased the sensitivity of the diagnosis and have become an accepted standard for the diagnosis of patient *O. volvulus* infection [107]. However, obtaining biopsy is a painful process and carries some risk of transmitting blood-born infections, leading to community resistance [107]. Therefore, skin snip-based assays are not recommended as a primary diagnostic for the verification of *O. volvulus* elimination [107]. Recently, serological tests to detect IgG4 antibodies to the parasite antigen Ov-16 have been developed and used for the identification of incident infections in communities having already undergone MDA [107,108]. However, IgG4 response takes time to develop and therefore will not detect new infection. Novel, non-invasive molecular diagnosis, which can distinguish past or active infection, and can be used to monitor the progression of the disease is in demand.

*O. volvulus* miRNAs have been detected in the serum of affected individuals [109], raising the potential of using *O. volvulus* miRNAs as diagnostic biomarkers for the disease. However, a recent attempted test of a set of 17 parasitic miRNAs by quantitative (q)RT-PCR using the miRCURY Locked Nucleic Acid (LNA) universal RT microRNA PCR system (Exiqon, Denmark) showed disappointing inconsistent results [100]. Of the 17 parasitic miRNAs tested, only three miRNAs (bma-miR-236–1, ov-miR-100d, ov-bantam-a) were detected and in only a few samples; they are not universally present in all or most infected individuals, which is an essential property for use as a diagnostic marker [100].

In spite of this negative report, the principle of plasma-derived parasitic miRNAs as diagnostic biomarkers for river blindness is highly plausible. With the genome of *O. volvulus* being deciphered [104], more sensitive detection systems should be tested to fully explore the potential of using parasitic miRNAs as non-invasive diagnostic biomarkers for river blindness.

### 1.4. miRNAs in Fungal Keratitis

Fungal keratitis (FK) is frequently caused by filamentous fungi (*Fusarium*, *Aspergillus*, *Phaeohyphomycetes,* and *Scedosporium apiospermum*) and yeast-like fungi (*Candida albicans* and other *Candida* species). FK is characterized by rapid progression with corneal ulceration and a stromal inflammatory infiltrate [110,111,112]. Although treatment with the polyene, natamycin, and amphotericin B have been the mainstay of management of disease, 15–27% of patients with fungal keratitis require surgical intervention, which often has relatively poor prognosis [111,113]. Alternative, improved treatment is in demand.

Recent reports suggest that miRNAs are involved in fungal keratitis. Deep RNA sequencing in the corneas of five FK patients with culture positive for *Aspergillus flavus* and three normal controls detected that 75 miRNAs were differentially expressed between infected and normal control corneas [111]. Out of the highly differentially expressed (>6 folds), 16 were further validated by quantitative (q)RT-PCR, as seen in Table 1. Functional annotation of these highly dysregulated miRNAs, including miR-511–5p, miR-142–3p, miR-155–5p, miR-451a, suggested regulation on inflammation and the wound healing processes [111]. Among these, increased expression of miR-451a in keratitis appeared to correlate with reduced expression of one of its target genes, macrophage migration inhibitory factor (MIF), suggesting potential regulatory functions [111]. Further study to confirm whether miR-451a targets MIF in a specific cell type and how this regulation modulates the pathogenesis of fungal keratitis will provide more insight into the role of miR-451a in fungal keratitis and its potential as a therapeutic target.

### 1.5. miRNAs in Bacterial Keratitis

Bacterial keratitis is most frequently associated with complications of extended contact-lens usage in the industrialized countries. *Pseudomonas aeruginosa* (PA), a Gram-negative bacterium and an important human pathogen, remains the most commonly recovered causative organism in contact lens-related keratitis in developed countries, and one of the most rapidly developing and destructive blinding diseases of the cornea [114]. PA infection of the cornea induces inflammatory epithelial edema, stromal infiltration, corneal tissue destruction, ulceration, scarring, reduced visual acuity, and occasionally vision loss [114]. Monotherapy with fourth-generation fluoroquinolones (moxifloxacin or gatifloxacin) is the most commonly used treatment [114,115]. However, PA has tremendous ability to develop resistance to multiple antibiotics and has consequently joined the ranks of “superbugs” [116]. In addition, eliminating the pathogen from the cornea constitutes only a part of the treatment of PA keratitis, as host excessive, uncontrolled immune/inflammatory responses play a major role in the corneal pathology and severity of PA keratitis. Therefore, novel therapies targeting both the pathogen and host immune responses are in demand for efficient management of the disease.

Several pioneering studies have provided the first insights into the roles of miRNAs in the pathogenesis of bacterial keratitis through modulating the functions of multiple cell types involved in the disease, as seen in Table 1. Mun J et al. first investigated the roles of miRNAs in human corneal epithelial cell (CEC) in response to PA infection [117]. Tear fluid protects ocular surface epithelial cells against bacterial virulence [118,119]. Tear fluid-induced upregulation of epithelial-derived innate defense genes, including RNase 7 and ST2 (Il1rl1), both of which reduces bacterial internalization by CECs [120,121,122,123], contributes to the protective effect. Mun et al. employed miRNA expression profiling in human CECs (HCECs) treated with PA antigen with or without human tear fluid [117] and identified that miR-762 and miR-1207 [124,125,126] were upregulated, while miR-92 [127,128,129] and let-7b [130,131] were downregulated in HCECs treated with tear plus PA when compared with those treated with PA alone [117], suggesting their potential roles in modulating tear-induced gene-expression changes upon PA infection, as seen in Table 1. Intriguingly, in vitro data showed miR-762 appeared to target and downregulate RNase 7 and ST2 [117], while tear fluid alone simultaneously increased the expression of miR-762 [117], and RNase 7 and ST2 [120,121,122,123], suggesting that miR-762 may not regulate these tear-induced innate defense genes in vivo; alternatively, other unknown mechanisms counteract the inhibitory effects of miR-762 on RNase 7 and ST2 [117]. Therefore, the roles of miRNAs in tear-induced increased expression of innate defense genes in corneal epithelial cells remain to be further defined.

Yang K et al. reported that miR-155 is induced in the cornea of both human and mouse after PA infection as early as one day post-infection (dpi). miR-155 appeared to be predominantly expressed in macrophages (Mϕ) compared to neutrophils, and was drastically induced upon PA infection [132]. miR-155 enhanced bacterial burden and promoted corneal susceptibility to PA keratitis; in contrast, inactivation of miR-155 in mice resulted in a reduced bacterial load and decreased severity of experimental PA keratitis [132]. Mechanistically, Yang et al. provided evidence that miR-155 inhibited macrophage-mediated phagocytosis and intracellular bacterial killing through modulating the expression of induced nitric oxide synthase (iNOS) and the production of nitric oxide (NO) [132]. This effect is possibly mediated by targeting Rheb, a gene known to interact with mammalian Target Of Rapamycin (mTOR) [133] and increase mTOR activity [134]. In contrast, inhibition of mTOR by treatment with rapamycin was shown to increase bacterial burden and promote PA keratitis and reduce neutrophil bactericidal activity in response to PA infection [135]. These data suggest that miR-155 could be a therapeutic target for treatment of PA. Knockdown of miR-155 may increase the phagocytosis and intracellular killing capacity of macrophages and help eliminate the bacteria from the infected cornea.

Recently, with both in vivo and in vitro approaches, we showed that the conserved, paralogous miRNA cluster, the miR-183/96/182 cluster (miR-183/96/182), modulates the corneal response to PA infection through its regulation of pathogenesis of the disease at multiple levels [136,137,138,139]. Initially, we and others identified miR-183/96/182 as a sensory organ-specific miRNA cluster, as it is highly specifically expressed in all major sensory organs [140,141,142] and is required for normal development and functions of all major sensory domains [53,140,143,144,145,146,147]. Point mutations in the seed sequence of miR-96 resulted in non-syndromic hearing loss in both human and mouse [43,48]; inactivation of miR-183/96/182 in mice resulted in multi-sensory defects [53,143,144]. Since the cornea is one of most heavily sensory innervated tissues, we first hypothesized that miR-183/96/182 modulates the corneal response to PA infection through its regulation of sensory innervation and neuroimmune/inflammation of the cornea. As expected, we showed that miR-183/96/182 is expressed in the trigeminal ganglion (TG) and the cornea; inactivation of miR-183/96/182 in mice resulted in decreased corneal nerve density in the subbasal plexus and the expression of nociceptor transient receptor potential vanilloid 1 (TRPV1) and multiple pro-inflammatory neuropeptides, including Tac1 [the precursor gene for substance P (sP)], calcitonin gene-related peptide (CGRP), and chemokine (C-X3-C motif) ligand 1 (Cx3cl1) [136]. Intriguingly, miR-183/96/182 knockout (ko) mice showed a significantly decreased corneal inflammatory response to PA infection and reduced severity of PA keratitis [136]. There were decreased infiltrating neutrophils and slightly increased bacterial load in the ko mice at 1 dpi. However, at 5 dpi, the bacterial load in the ko mice was decreased when compared with their wild-type controls [136]. Furthermore, we demonstrated that miR-183/96/182 is also expressed in innate immune cells, including Mϕ and neutrophils [136]. Inactivation or knockdown of miR-183/96/182 resulted in increased phagocytosis and intracellular bacterial killing capacity of both Mϕ and neutrophils [136]. This effect could be the result of increased production of reactive nitrogen species (RNS) and reactive oxygen species (ROS) in these innate immune cells through its regulation of Nox2, a key enzyme required for the generation of superoxide (O_2_^−^) and other microbicidal ROS and RNS, as seen in Table 1 [137,148,149]. In addition, we showed that miR-183/96/182 promotes the production of pro-inflammatory cytokines in Mϕ through targeting DAP12 [137], which is consistent with the observation of an overall decreased level of pro-inflammatory cytokines in the cornea in response to PA infection [136].

Moreover, our collaborators showed that miR-183/96/182 is not only expressed in innate immune cells, it is also one of the highest induced miRNAs during helper T (Th)-17 cell differentiation, and promotes Th17 cell pathogenicity by negatively regulating the expression of transcription factor Foxo1—a negative regulator of IL-1R1, as seen in Table 1 [138]. Inactivation of the miR-183/96/182 cluster in mice resulted in a decreased pathogenicity of Th17 cells and a reduced production of pro-inflammatory cytokines, leading to a decreased severity of Experimental Autoimmune Encephalomyelitis (EAE) [138]. Since IL-17 activity and Th17 have been shown to promote neutrophil infiltration and the severity of PA keratitis [150,151], we predict that miR-183/96/182′s regulation of Th17 pathogenesis also contributes to its overall effect on PA keratitis. Collectively, these data suggest that the miR-183/96/182 cluster modulates the corneal response to PA infection and the resolution of PA keratitis at multiple levels, including sensory innervation and neuroimmune/inflammation as well as both the innate and adaptive immunity. These data also indicate that miR-183/96/182 is a potential therapeutic target for the treatment of PA keratitis. Knockdown of the function of the miR-183/96/182 cluster is predicted to have concerted therapeutic effects by enhancing phagocytosis and intracellular killing capacity of Mϕ and neutrophils, decreasing the production of pro-inflammatory neuropeptide by sensory nerve endings, and pro-inflammatory cytokines/chemokines by Mϕ and Th17 cells, resulting in controlled infiltration of neutrophils to avoid excessive collateral tissue damages.

*Staphylococcus aureus* (*S. aureus*) is a leading cause of keratitis worldwide [85]. However, there is no report on roles of miRNAs in *S. aureus* keratitis and keratitis caused by other bacteria (based on a Pubmed search with keywords of *Staphylococcus aureus*, keratitis, miRNA on 20 August 2019). This search result itself reflects an existing knowledge gap in miRNAs in bacterial keratitis.

### 1.6. miRNAs in Viral Keratitis

Herpes simplex virus is an important human pathogen causing diseases worldwide. The majority of the world’s population has been infected with HSV-1 but in a latent state [152]. In addition to oral and genital lesions, HSV-1 can cause ocular diseases in all tissues of the eye. HSV-1 induced herpes stromal keratitis (HSK) is the most frequent viral eye infection in developed countries and the leading cause of corneal blindness due to an infectious agent [153,154]. The primary ocular HSV-1 infection rarely causes clinical symptoms; it occurs early in life with epithelial keratitis, including punctuate vesicular eruptions as well as dendritic-shaped and geographic lesions. These lesions are caused by viral replications in the epithelial cells, which destroy the cells, referred to as a viral cytopathic effect [152]. The primary herpes epithelial keratitis (HEK) is usually self-limiting, but heals more rapidly when treated with antiviral drug [152,155,156]. HSV-1 is cleared from the cornea during the primary infection largely by the innate immune response. However, during the primary infection of the cornea and other oral facial regions, the virus gains access to the termini of sensory neurons of the trigeminal ganglia (TG) and is transported through retrograde axonal transport to the cell body of the TG to establish a state of latency; the viral DNA is retained as a circular episomal DNA in the neuronal nuclei while no infectious virus particles are produced [152,157]. More severe HSK is often the result of the recurrent herpetic disease because of HSV-1 reactivation from the latency; however, it can occur as a progression of HEK or the primary manifestation of keratitis [152,158]. Major clinical signs of HSK include stromal opacity, edema, neovascularization, and shedding into the cornea. Repeated recurrence of HSK can lead to progressive, irreversible corneal scarring and blindness [152]. The potent host immune response to viral proteins plays a major role in damage to the cornea. Therefore, prevention of the reactivation of the latency and controlling the host immune/inflammatory response are essential to the treatment of HSK. Current treatment for HSK includes topical and oral administration of acyclovir, ganciclovir, trifluothymidine, penciclovir, and valacyclovir [159]. Topical corticosteroid application also helps reduce stromal inflammation [159]. However, these treatments only reduce the severity of lesion and control further viral spread, but does not provide a cure. Acyclovir and its derivatives prevents viral replication by inhibiting viral DNA elongation; it only affects newly synthesized viral DNA, but it does not eliminate existing viral DNA from infected cells [159]. In addition, long-term use of acyclovir and its derivative results in resistance; recurrence can still occur [159,160,161]. Long-term topical use of corticosteroid also has its own ocular complications, including cataract, glaucoma, and opportunistic microbial infection [162]. Therefore, there is a pressing need to develop novel therapeutic strategy for treatment of HSK.

The role of miRNAs in HSK is one of the most studied areas of miRNAs in ocular infection. miR-155 has been shown to play an important role in HSK through its regulation on the immune system. HSV-1 infection of mouse cornea resulted in increasing upregulation of miR-155 in the cornea at 2, 7, and 15 dpi [163]. This upregulation occurred mainly in Mϕ and CD4+ T cells, especially in activated CD4+T cells, and, to a lesser extent, in neutrophils in the infected cornea [163]. miR-155 ko mice showed decreased severity and angiogenesis in HSK, accompanied by significantly reduced infiltrating CD4+ T cells and Th1 and Th17 responses in both the infected cornea and the lymphoid organs, including the draining lymph nodes (DLNs) and the spleen [163]. The decreased number of infiltrating CD4+ T cells is possibly a result of decreased proliferation of these cells in the cornea and DLNs after HSV-1 infection, suggesting that miR-155 promotes CD4+ T cell proliferation [163]. In vivo silencing of miR-155 by conjunctival injection of antigomir-155 nanoparticles at preclinical (1 dpi) and early clinical stages (5 dpi) resulted in decreased severity of HSK with decreased infiltration of CD4+ T cells and neutrophils as well as reduced production of pro-inflammatory cytokines, including IL-1β, IL-6, IFN-γ, and IL-17, and chemokines, e.g., Cxcl-1 and Ccl-2 [163]. Th1 responsiveness is a major player to orchestrate HSK [164]. Ship1 and IFN-γΡα, which are known to be targets of miR-155 and play important roles in regulating IFN-γ expression and Th1 differentiation [165,166], were significantly increased in activated CD4+ T cells of the DLNs of miR-155 ko mice [163], suggesting that the function of miR-155 in HSK is, at least partially, mediated by its regulation of Th1 cell development through targeting Ship1 and IFN-γRα. These data suggest that miR-155 is a potential target for drug development to control HSK.

The cornea is an avascular tissue; the transparency of the cornea is requisite for normal vision. Corneal neovascularization during HSK allows immune/inflammatory cells to gain access to the cornea, contributing to the severity of HSK and visual impairment [167,168]. Prevention and suppression of neovascularization is one of the major objectives of treatment of HSK [168]. In this regard, miR-132, known to activate the endothelium and facilitate pathological angiogenesis by targeting p120RasGAP, a negative regulator of angiogenic Ras activity [169], was shown to be upregulated in the cornea after HSV-1 infection [170]. Both VEGF-A and IL-17a regulate the expression of miR-132 and contribute to its upregulation in the cornea in response to HSV-1 infection [170]. In vivo silencing of miR-132 by subconjunctival injection of antigomiR-132 nanoparticles in HSV-infected mice reduced corneal neovascularization and the severity of HSK, while the number of infiltrating CD4+ T cells and neutrophils in the cornea was significantly decreased [170]. The anti-angiogenic effect of antigomiR-132 was accompanied by an increased expression of p120RasGAP and reduced Ras activity in the endothelial cells of the cornea [170], suggesting that p120RasGAP mediates miR-132′s pro-neovascularization function in this context; knockdown of miR-132 is a potential therapeutic strategy for HSK treatment.

As discussed above, after primary ocular infection, HSV-1 ascends through axons and persists throughout life as a latent infection in sensory neurons of TG; reactivation of the virus causes recurrent HSK leading to visual impairment and blindness [154]. Recent reports suggest that miRNAs derived from both HSV-1 and the host genomes play important roles in the latency-reactivation cycle. Establishment and maintenance of HSV latency requires host survival and entails repression of productive-cycle (“lytic”) viral gene expression [171]. The latency-associated transcript (LAT) of a HSV-1 gene is critical to the HSV-1 latency-reactivation cycle through its anti-apoptosis activity and immune evasion properties [154,172,173,174,175]. Recently, eight LAT-associated miRNAs (miRs H1–H8) have been identified in and near the LAT locus of the HSV-1 genome [176,177]. miR-H2 is expressed in the LAT direction and overlaps part of a major exon of the HSV-1 ICP0 gene, an immediate early gene which trans-activates lytic gene expression [171,172,173], but in an antisense orientation [154,172,173]. Recent reports showed that miR-H2 targets ICP0 and promotes latency of HSV-1. Disruption of miR-H2 without altering the amino acid sequence of the ICP0 gene resulted in increased production of ICP0 in host cells, and enhanced neurovirulence in mice after ocular infection of HSV-1 (primary HSK) and more rapid reactivation [154,178]. Simultaneous inactivation of LAT transcript and miR-H2 resulted in more robust reaction [178]. These data suggest that HSV-1-derived miR-H2 modulates HSV-1 neurovirulence and reactivation through targeting ICP0.

Intriguingly, a host neuron-specific miRNA, miR-138, is also shown to target and repress the expression of ICP0 and other lytic genes of HSV-1. A mutant HSV-1 (M138) with disrupted miR-138 target sites in ICP0 mRNA exhibited increased expression of ICP0 and other lytic proteins in infected neuronal cells in culture [171]. Consistently, in vivo, post-corneal inoculation, M138-infected mice showed higher expression of ICP0 and other lytic transcripts in the TG during latency establishment, and exhibited increased mortality and encephalitis symptoms [171]. After full establishment of latency, an increased fraction of TG neurons expressed lytic transcripts in M138-infected mice. These data suggest that miR-138, a neuronal miRNA, represses HSV-1 lytic gene expression and promotes host survival and viral latency [171]. Therefore, miR-138 is another potential therapeutic target to develop new treatment against reactivation of HSV-1 virus, promoting latency.

For further detailed information on miRNAs in HSK as well as HSV infection in other tissues, please refer to previous reviews by Mulik S et al. (2013) [179] as well as Bhela S and Rouse BT (2017) [180]. A literature search on PubMed on 20 August 2019 using the keywords “viral, keratitis, miRNA” did not recover additional reports on miRNAs in viral keratitis other than HSK. This search result reveals the shortage in our current knowledge on miRNAs in ocular infection and calls for more research effort in the future.

## 2. Discussion and Conclusions

Although miRNA research in ocular infection remains in its infancy, the handful of pioneering studies summarized in this review have demonstrated important roles of miRNAs in the pathogenesis of ocular infection and their potential as therapeutic targets and diagnostic biomarkers. One of the most prominent findings is that miRNAs modulate the ocular infection through its regulation on innate and adaptive immunity. In PA keratitis, both miR-155 [132] and the miR-183/96/182 cluster [136,137] have been shown to suppress phagocytosis and intracellular bacterial killing of PA by neutrophils and Mϕ. Knockout or knock-down of the function of miR-155 and the miR-183/96/182 cluster resulted in increased phagocytosis and bacterial killing, which contributes to the decreased severity of PA keratitis in the ko mice. Interestingly, these effects of both miR-155 and the miR-183/96/182 cluster appeared to be achieved by modulation of ROS production albeit through different targets and pathways. For miR-155, Yang et al. showed that Rheb, which can directly interact with mTOR and increase mTOR activity [133,134], is targeted and mediates the regulation of miR-155 on bactericidal capacity of innate immune cells [135]. In contrast, for the miR-183/96/182 cluster, Nox2, one of the key enzymes required to generate superoxide and other bactericidal ROS and RNS, is targeted by miR-182 and 96, contributing to its effect on phagocytosis and bacterial killing by Mϕ and neutrophils [136,137].

In HSK, Bhela et al. demonstrated that miR-155 modulate the pathogenesis through its regulation on adaptive immunity, specifically, CD4+ T cells proliferation and Th1 and Th17 responses to HSV-1 infection [163] by targeting Ship1 and IFN-γRα, both of which are involved in IFN-γ expression and Th1 differentiation [165,166]. These reports on miR-155 [132,163] exemplify that one miRNA can be involved in the pathogenesis of different ocular infectious diseases through its roles in different domains of immunity. One of the major players in immunity, miR-155, is also dysregulated in other ocular infection, e.g., trochomatous follicular inflammation [88] and fungal keratitis [111]. Further studies on miR-155 in these ocular infectious diseases are warranted to gain deeper insights into its pathological roles.

Another important finding illustrated that one miRNA can simultaneously affect different aspects of the pathogenesis of an ocular infectious disease and, in concert, impose significant impact on the development of the disease. In this regard, the miR-183/96/182 cluster modulates PA keratitis not only through its regulation on innate immunity of the cornea but also through its regulation on sensory innervation, neuro-immune/neuro-inflammation [136] and, possibly, adaptive immunity, specifically, Th17 pathogenicity [138].

Current studies also demonstrate that miRNAs involved in different aspects of the pathogenesis of an ocular infection all could have significant impact on the disease. For example, although miR-155 modulates HSK through its regulation on CD4+ T cell proliferation and Th1 and Th17 responses to HSV-1 infection [163], Mulik et al. demonstrated that pro-angiogenic miRNA miR-132 promotes neovascularization of the cornea in response to HSV-1 infection [170] and also contributes to the development of HSK. It would be interesting to explore whether simultaneous manipulation of miR-132 in vascular endothelial cells and miR-155 in CD4+ T cells could have an additive or synergistic effect on the pathogenesis of HSK and its treatment.

In addition to the miRNAs which have direct involvement in ocular infectious diseases and are summarized in this review, many other miRNAs have been shown to play important roles in dry eyes [181,182] and other autoimmune diseases [183,184,185,186], in innate and adaptive immunity and inflammation [187,188,189,190,191,192,193], in neuroimmune/neuroinflammation [194,195,196], in microbial infection in other systems [84,197,198], as well as in angiogenesis [199,200] and lymphangiogenesis [201,202]. We predict that it will be shown that many of these miRNAs also contribute to the pathogenesis of various ocular infectious diseases. However, a miRNA known to be involved in inflammation, immunity, angiogenesis, and lymphangiogenesis in other organ systems or in different physiological and pathological conditions in the eye may have different functions in ocular infectious diseases; their exact roles in ocular infection must be vigorously tested experimentally in ocular infectious disease models in vitro and in vivo.

Several miRNAs have been shown to be potential therapeutic targets for treatment of ocular diseases. In vivo silencing of miR-155 by antagomirs-155 nanoparticles delivered to HSV-1 infected mouse cornea resulted in diminished HSK lesion and neovascularization [163]. Knockdown of pro-angiogenic miR-132 in mouse cornea also led to decreased neovascularization and decreased severity of HSK [170]. Our unpublished data (Xu and Hazlett et al.) suggested that topical application of LNA-anti-miR-183/96/182 decreased the inflammatory response of the cornea to PA infection and decreased the severity of PA keratitis. Since knockdown or knockout of both miR-155 and miR-183/96/182 enhanced the capacity of phagocytosis and intracellular bacterial killing by Mϕ and/or neutrophils [132,136,137], it is reasonable to speculate that simultaneous silencing of miR-155 and the miR-183/96/182 cluster may have an additive or synergistic effect boasting their bactericidal power. This enhancement on innate immunity may have important implications in management of infectious diseases, as it could provide an alternative strategy to combat multidrug resistant strains of bacteria.

Targeting a miRNA as a therapy involves either enhancing or inhibiting its functions based on its roles in the pathogenesis of the disease. Using synthetic ORNs that mimic the native miRNA duplex to deliver miRNAs and enhance its function have been widely used in in vitro assays [58,59,136,137,203] and in vivo trials [204]. Other strategies to enhance the function of miRNAs include deliveries by lentivirus [60,205], adenovirus [206], and adeno-associated viruses [63]. To inhibit the functions of miRNAs, the most popular approach is transfection of single-stranded synthetic anti-sense ORN to sequester endogenous miRNAs [136,137]. Other approaches include vector-based delivery of transcripts containing multiple artificial miRNA-binding sites, which act as decoy or miRNA sponges [207]. Various chemical modifications of ORNs, e.g., 2′-OMe, 2′-MOE, 2′-fluoro substitution of the 2′-OH of the ribose sugar or LNA modification of the backbone structure, have significantly increased their stability in vitro and in vivo [56,61,62,65,208,209]. Recent advances in nanotechnology-based delivery systems have significantly enhanced the stability of ORNs, cellular accessibility, and tissue targeting [210,211,212], some of which are biodegradable and safer for in vivo delivery [213,214].

In spite of progress, major challenges to miRNA-based therapy still exist, e.g., tissue- or cell type-specific targeting, off-target effects, and safety [56,215]. miRNAs function in a cell-type specific fashion; cell type-specific delivery is the key to achieve therapeutic effect without unintended off-target functional consequences from non-specific delivery of a miRNA mimic or anti-miR into other cell types. Virus-based delivery has the potential to confer tissue-specific expression when engineered under the control of tissue-specific promoters; however, viral carriers often have their own disadvantages, such as causing systemic toxicity or an immune response [216,217]. miRNAs in the same miRNA family have high sequence homology; under current design, anti-miRs are generally unable to distinguish between miRNAs within the same family, leading to non-specific, off-target, promiscuous inhibition [215]. In addition, anti-miRs and their carrier proteins may be detected by the host immune system and cause adverse immune/inflammatory responses. Pattern recognition receptor, Toll-like receptor (TLR)3 on innate immune cells recognizes dsRNA and can be activated by siRNAs in a sequence-independent manner [218,219]; in contrast, TLR7 and TLR8 recognize single strand RNA sequences and evoke IFNα mediated inflammatory response [220,221,222,223]. Some of the chemically modified ORNs have been shown to induce sequence-independent toxicity in vivo, of which the most common effects are inhibition of coagulation, activation of complement cascade, and immune cell activation [224,225]. Liver toxicity is another outstanding concern for systemic application of chemically modified ORNs [215,224,226,227]. Therefore, application of miRNA-based therapy must be vigorously tested in vitro and in vivo to avoid off-target effects and unintended toxicity. For more information on the current status, the pros and cons of miRNAs therapeutics, please refer to the many reviews on this topic [56,214,217,228].

In this regard, miRNA-based therapy for ocular infectious diseases may have special advantages because of the relatively easy access to the ocular surface by topical application and the interior of the eye by intraocular injection without systemic administration. However, careful characterization must be made to evaluate the impact of any miRNA-based therapy on the overall functions of the eye and ocular immune/inflammatory response in addition to its intended therapeutic effects.

In conclusion, miRNAs are proven to be important fine-tuners of gene expression regulation. Several pioneering studies have demonstrated that miRNAs play important roles in the pathogenesis of ocular infection by modulating several major, interconnected systems, including the innate and adaptive immunity, sensory innervation and neuroimmune/neuroinflammation, angiogenesis, and neovascularization. The field of miRNAs in ocular infection is still largely unexplored territory. We predict that future research on miRNAs in ocular infectious diseases will be especially fruitful in the coming years and will provide unprecedented new insights into molecular mechanisms of miRNAs in the pathogenesis of various ocular infectious diseases; it will identify previously unrecognized therapeutic targets for new drug development allowing quantitative, stage-specific, and cell-type specific therapy, and diagnostic biomarkers for ocular infection.

## Figures and Tables

**Figure 1 microorganisms-07-00359-f001:**
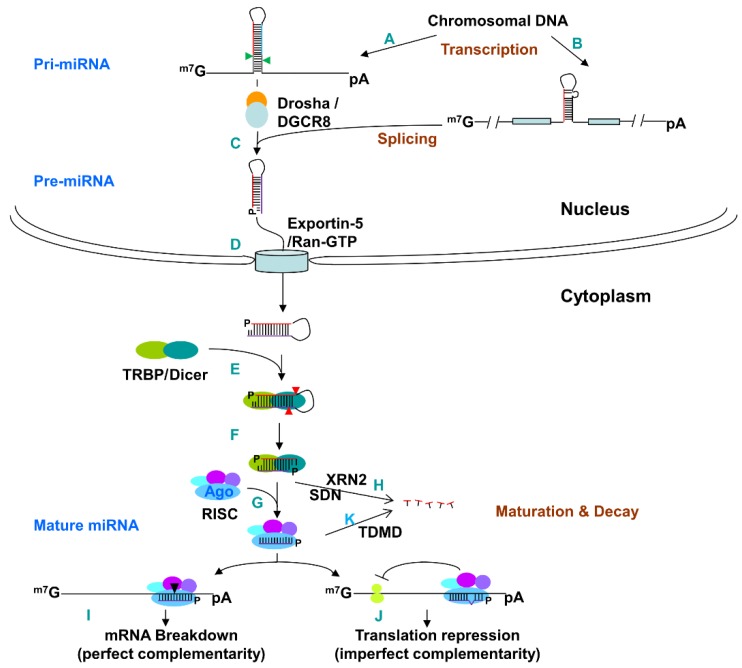
miRNA biogenesis and functions. In the nucleus, the primary transcript of miRNA, referred as pri-miRNA, is mostly transcribed by RNA polymerase II (**A**). More than 25% of the conserved and >50% of the poorly conserved miRNAs are derived from introns of protein-coding genes (**B**). The pri-miRNAs fold into hair-pin structures, which are cleaved by an RNase III endonuclease, Drosha in the Drosha-DGCR8 complex, to form 60–70 nt stem loop intermediates, known as pre-miRNAs, with a 2-nt 3′ overhang (**C**). Pre-miRNAs are transported to the cytoplasm by Ran-GTP and an export receptor, Exportin 5 (**D**). In the cytosol, pre-miRNAs are cleaved by another RNase III endonuclease, Dicer in the Dicer/TRBP complex, to form ~22-bp miRNA duplexes with a 2-nt 3′ overhang (**E,F**). TRBP recruits Agonaute protein Ago2 and other Ago proteins to initiate the assembly of the RNA-induced silencing complex (RISC). One strand of the duplex becomes mature miRNA and is incorporated in the RISC complex (**G**); the other strand, miRNA*, is degraded by small RNA degrading nuclease (SDN) (in *Arabidopsis*) and 5′-3′ exonuclease 2 homolog (XRN2) in *Caenorhabditis elegans* (**H**). Mature miRNAs base pair with their target mRNAs in the 3′ UTR (**I,J**). When the miRNA and the target sites have perfect or nearly perfect complementarity, miRNAs direct cleavage of the target mRNAs by Ago2 (**I**); when the base-pairing is imperfect, miRNA with the RISC can destabilize the mRNA by deadenylation and subsequent decapping, and repress translation of the targeted mRNA by blocking translation initiation and/or inhibiting elongation (**J**). (Modified from Xu, 2009 [6]). Muture miRNA can be also degraded through target RNA-directed miRNA degradation (TDMD) (**K**).

**Table 1 microorganisms-07-00359-t001:** miRNAs Involved in Ocular Infection.

Disease / Stage	miRNA	Species	Tissue / Cell Type	Validated Changes	Potential Targets*	Experimentally Confirmed	Functions and/or Pathways Involved	Reference
Trochoma/Follicular	miR-155–5p	human	conjunctival swabs	up	not tested in TF	na	hematopoisis, immune cells development and function	[88,91]
	miR-150–5p	human	conjunctival swabs	up			hematopoietic cells	[88,92]
	miR-142–5p	human	conjunctival swabs	up			hematopoietic cells	[88,92]
	miR-181a/b-5p	human	conjunctival swabs	up			hematoposis, inflammation	[88]
	miR-342–3p	human	conjunctival swabs	up			cell proliferation, inflammation	[88,93,94]
	miR-132–3p	human	conjunctival swabs	up			hematopoiesis, inflammation	[88]
	miR-4728	human	conjunctival swabs	down			focal adhesion and wound healing, cancer	[88,95]
	miR-184	human	conjunctival swabs	down			Corneal development and function, wound healing, ischemia-induced neovascularization	[88,93,94,95]
Trochoma/Scarring	miR-147b	human	conjunctival swabs	up	not tested in TS	na	fibrosis and epithelial cell differentiation	[89]
	miR-1285	human	conjunctival swabs	up				
Fungal keratitis	miR-511–5p	human	cornea	up	not tested in FK	na	Immune response, cell proliferation, tumor suppression	[111]
	miR-451a	human	cornea	up			Cell proliferation, migration	
	miR-223–3p	human	cornea	up			Cell proliferation, cell invasion, and migration, apoptosis, wound inflammation	
	miR-21–5p	human	cornea	up			Cell proliferation, cell cycle, apoptosis, wound inflammation	
	miR-142–5p	human	cornea	up			Cell proliferation, apoptosis	
	miR-142–3p	human	cornea	up			Cell viability, proinflammatory, signaling	
	miR-618	human	cornea	up			Apoptosis, invasion, migration	
	miR-155–5p	human	cornea	up			Oncomir, immune responses, wound inflammation	
	miR-144–5p	human	cornea	up			Cell proliferation	
	miR-144–3p	human	cornea	up			Proinflammatory response	
	miR-146a-5p	human	cornea	up			Inflammation, cell migration, invasion, wound healing	
	miR-146b-5p	human	cornea	up			Inflammation, cell migration, invasion	
	miR-424–5p	human	cornea	up			Wound healing	
	miR-124–3p	human	cornea	down			Cell proliferation, apoptosis	
	miR-204–5p	human	cornea	down			Wound healing	
	miR-184	human	cornea	down			Cell proliferation, migration, wound healing	
Pseudomonas aeruginosa	miR-762	human	corneal epithelial cell line	up	RNase 7, ST2, Rab5a	Yes	bacterial internalization	[117,124,125,126,127,128,129,130,131]
	miR-1207	human	corneal epithelial cell line	up	not tested in PA keratitis	na	Ovarian cancer, nasopharyngeal cancer, pancreatic cancer, mesenchymal stromal cell expansion	
	miR-92a-3p	human	corneal epithelial cell line	down	not tested in PA keratitis	na	hematoposis, immune cells, cancer	
	let-7b-5p	human	corneal epithelial cell line	down	not tested in PA keratitis	na	cell cycle, cancer, wound healing	
	miR-155–5p	human/ mouse	human and mouse cornea; mouse peritoneal macrophages, and cell line, RAW264.7	up	Rheb	yes	Macrophage phagocytosis and intracellular killing; ROS production	[132]
	miR-183/96/182 cluster	mouse	cornea, peritoneal macrophage and neutrophils, macrophage cell line Raw264.7, Th17 cells	up	Nox2, DAP12, Foxo1	yes	Macrophage and neutrophil phagocytosis and intracellular killing; ROS production; cytokine production; Th17 pathogenecity;	[136,137,138]
HSK	miR-155–5p	mouse	cornea, DLN, spleen, CD4+ T cells	up	Ship1 and IFN-gRa	yes	CD4+ T cell proliferation, Th1 differentiation, IFNg expression	[163]
	miR-132–3p	mouse	cornea, corneal endothelial cells	up	p120RasGAP	yes	angiogenesis	[170]
	miR-H2	rabbit	skin cells	up	ICP0	yes	decrease neurovirulence and reactivation, promote latency	[154,178]
		mouse	eye swabs, TG					
		human	neuroblastoma cell line SY5Y					
		monkey	kidney fibroblast cell line CV-1					
	miR-138–5p	mouse	neuronal cell line, Neuro-2A, eye swab, TG	unknown	ICP0	yes	decrease neurovirulence and reactivation, promote latency	[171]

*: only the targets and functions experimentally tested in these studies are listed.

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
