# Peer review of "MicroRNAs in Ocular Infection"

_microorganisms, 2019, doi:10.3390/microorganisms7090359_

Round 1
Reviewer 1 Report
Biogenesis and mode of action of miRNAs is described extensively in many research and review articles, along with graphical illustrations. To make it easier for would be readers, authors should provide a schematic illustration of these processes. If authors cannot draw one on their own, they should at least cite the different articles with this type of illustration. Since miRNA research in ocular infection is still in its infancy as stated by the authors, authors should briefly review examples of studies where miRNAs have been targeted for therapy, and or used as biomarkers for disease diagnosis, illustrating how these findings would translate to ocular infections Authors should clearly justify the purpose of current review article under subheading 1.1. and align the discussion to this goal To increase depth and substance of the current review article, authors should provide a brief review of the current methods for diagnosis of ocular infections, treatments and prognosis, highlighting any loopholes There is a growing number of review articles describing the role of miRNAs for example in dry eye, and cornea neovascularization. Authors should describe how the current review fits in the broader topic of miRNA in the eye/cornea, while citing the relevant literature, given that infection of the cornea may lead to inflammation, and neovascularization in this tissue To provide a more comprehensive review, and to match the broad nature of the title of the manuscript, authors should discuss bacterial keratitis resulting from Staphylococcus aureus, and viral keratitis other than HSK A table summarizing infections, miRNAs associated with them, together with the corresponding references should be included. Columns classifying the identified miRNA as either potential therapeutic targets and or biomarkers should also be included in the table Authors conclude that ‘’many of the miRNAs involved in ocular infection could be targets for new drug development allowing quantitative, stage-specific and cell-type specific therapy’’. I believe that the authors are aware of the potential hurdles in the journey leading to this, and as such, a brief description of the different methods used to modify miRNA expression should be provided, and the limitations of this type treatment be discussedAuthor Response
Reviewer 1
1. Biogenesis and mode of action of miRNAs is described extensively in many research and review articles, along with graphical illustrations. To make it easier for would be readers, authors should provide a schematic illustration of these processes. If authors cannot draw one on their own, they should at least cite the different articles with this type of illustration.
A: an illustration is provided as Fig.1.
2. Since miRNA research in ocular infection is still in its infancy as stated by the authors, authors should briefly review examples of studies where miRNAs have been targeted for therapy, and or used as biomarkers for disease diagnosis, illustrating how these findings would translate to ocular infections
A. We have added extensive descriptions on miRNAs as therapeutic targets and biomarkers on page 2 and 3, and all throughout the text, e.g. Page 6 in regard to miR-155 in PA infection, Page 7 in regard to miR-183/96/182 in PA infection, Page 10 and 11.
3. Authors should clearly justify the purpose of current review article under subheading 1.1. and align the discussion to this goal.
A. We have added in Section 1.1: This review will summarize what we have learned from these studies. It will also identify knowledge gaps in the field of miRNAs in ocular infection. We hope this review will provide reference knowledge and guidance for investigators new to this field.
We also revised the discussion to align with this goal.
4. To increase depth and substance of the current review article, authors should provide a brief review of the current methods for diagnosis of ocular infections, treatments and prognosis, highlighting any loopholes
A. We have added an overview of current treatment and diagnosis and challenges for the management of ocular infectious diseases in Section 1.1. And we also added brief reviews on current methods for diagnosis, treatment, and/or prognosis and challenges for each disease discussed in the following chapters.
5. There is a growing number of review articles describing the role of miRNAs for example in dry eye, and cornea neovascularization. Authors should describe how the current review fits in the broader topic of miRNA in the eye/cornea, while citing the relevant literature, given that infection of the cornea may lead to inflammation, and neovascularization in this tissue
A. We added extensive discussions on these miRNAs as well as miRNAs involved in autoimmunity, infection, lymphangiogenesis on Page 11.
6. To provide a more comprehensive review, and to match the broad nature of the title of the manuscript, authors should discuss bacterial keratitis resulting from Staphylococcus aureus, and viral keratitis other than HSK. A table summarizing infections, miRNAs associated with them, together with the corresponding references should be included. Columns classifying the identified miRNA as either potential therapeutic targets and or biomarkers should also be
included in the table
A: Currently, there is no report on roles of miRNAs in Staphylococcus aureus keratitis (Searched by Staphylococcus aureus, keratitis, miRNA on PubMed on 8/20/2019). Another search by “viral, keratitis, miRNA” did not recover any more reports on miRNAs in viral keratitis other than HSK reviewed in the current manuscript. A summary table on miRNAs involved in ocular infection with therapeutic potentials is added. For miRNAs as potential diagnostic biomarkers of ocular infection, the only report is mostly a negative result (Reference 97), and cannot draw any conclusion on whether these miRNAs can be used as biomarkers yet. Therefore, we did not include miRNAs as biomarkers for ocular
infectious diseases in the table.
7. Authors conclude that ‘’many of the miRNAs involved in ocular infection could be targets for new drug development allowing quantitative, stage-specific and cell-type specific therapy’’. I believe that the authors are aware of the potential hurdles in the journey leading to this, and as such, a brief description of the different methods used to modify miRNA expression should be provided, and the limitations of this type treatment be discussed
A: We have added descriptions and discussion on different methods to modify miRNA
expression and the challenges to miRNA-based therapy on Page 11.
Reviewer 2 Report
This review by Xu and Hazlett is a comprehensive discussion of the current literature regarding the roles of miRNAs in the pathogenesis of ocular diseases caused by a range of pathogens. It is well-written and well-organized. I have only a few comments that warrant further discussion before publication:
Major comments:
1. There needs to be clarification throughout sections that discuss specific diseases (1.2-1.6) of when the authors are referring to host or pathogen miRNAs e.g. in section 1.2, line 94, I assume the authors could add the word host to the sentence starting ‘Recent reports suggest that host miRNAs…’ which would substantially clarify the section.
2. It is well-known that host miRNAs play important roles in innate and adaptive immunity and can be modulated by many different pathogens beyond those that cause ocular disease e.g. influenza virus and Leishmania parasites. The authors discuss this to a limited extent for individual miRNAs/diseases and in the conclusion, but I think that a few sentences, or at the very least more citations, that broadly and conceptually discuss what is known about infection, immunity and miRNAs is warranted given the review content e.g. PMID 21842361 and 29656417.
3. Related to the above, the authors end section 1.2 with the sentence ‘These reports suggest that miRNAs are involved in the pathogenesis of trachoma. However, the exact mechanisms of their involvement are still to be uncovered.’ Could the authors expand on this e.g. do the authors have any hypotheses re. mechanisms based on what is known immune regulation? Do the pathways and targets described in the section suggest potential routes to pathogenesis?
4. In section 1.3, is there any evidence that pathogen and/or host miRNAs are involved in pathogenesis of river blindness, beyond being a potential biomarker?
Minor comments:
1. Line 30 ‘miRNAs constitute a new layer of gene-expression regulation at a post-transcriptional level’. Since they were first described in 1993 (PMID 24045890) I think it is misleading to use the word ‘new’-maybe ‘relatively recent’ is better?
2. Line 27 ‘increasing numbers of miRNAs have been identified in all species studied’. There are a number of well-studied pathogens e.g. the protozoan pathogens T. cruzi and Leishmania that appear to lack miRNAs and the RNAi machinery (PMID 15104593). Maybe ‘most’ is better to use here?
Author Response
Reviewer 2
This review by Xu and Hazlett is a comprehensive discussion of the current literature regarding the roles of miRNAs in the pathogenesis of ocular diseases caused by a range of pathogens. It is well-written and well-organized. I have only a few comments that warrant further discussion before publication:
There needs to be clarification throughout sections that discuss specific diseases (1.2-1.6) of when the authors are referring to host or pathogen miRNAs e.g. in section 1.2, line 94, I assume the authors could add the word host to the sentence starting ‘Recent reports suggest that hostmiRNAs…’ which would substantially clarify the section.
A: We agree with the reviewer, and made changes accordingly, now on line 143.
2. It is well-known that host miRNAs play important roles in innate and adaptive immunity and can be modulated by many different pathogens beyond those that cause ocular disease e.g. influenza virus and Leishmania parasites. The authors discuss this to a limited extent for individual miRNAs/diseases and in the conclusion, but I think that a few sentences, or at the very least more citations, that broadly and conceptually discuss what is known about infection, immunity and miRNAs is warranted given the review content e.g. PMID 21842361 and 29656417.
A. We have added extensive discussion with multiple references (Page 11), including the two examples provided by the reviewer (thanks!) to expand the coverage of miRNAs in innate and adaptive immunity, autoimmunity, angiogenesis and lymphangiogenesis.
3. Related to the above, the authors end section 1.2 with the sentence ‘These reports suggest that miRNAs are involved in the pathogenesis of trachoma. However, the exact mechanisms of their involvement are still to be uncovered.’ Could the authors expand on this e.g. do the authors have any hypotheses re. mechanisms based on what is known immune regulation? Do the pathways and targets described in the section suggest potential routes to pathogenesis?
A. We have expanded discussion as suggested (Page 4, Line 167-171). However, since little is known beyond that these miRNAs are dysregulated, we did not make more elaborate discussion as it will be purely speculative.
4. In section 1.3, is there any evidence that pathogen and/or host miRNAs are involved in pathogenesis of river blindness, beyond being a potential biomarker?
A: We are not aware of any evidence that pathogen and/or host miRNAs are involved in pathogenesis of river blindness yet, up to 8/10/2019 literature research.
Minor comments: 1. Line 30 ‘miRNAs constitute a new layer of gene-expression regulation at a post-transcriptional level’. Since they were first described in 1993 (PMID 24045890) I think it is misleading to use the word ‘new’-maybe ‘relatively recent’ is better?
A: We agree with the reviewer, and revised it accordingly (Line 30).
2. Line 27 ‘increasing numbers of miRNAs have been identified in all species studied’. There are a number of well-studied pathogens e.g. the protozoan pathogens T. cruzi and Leishmania that appear to lack miRNAs and the RNAi machinery (PMID 15104593). Maybe ‘most’ is better to use here?
A: Thank you for the careful review. It should be “…miRNAs have been identified in all metazoans studied”. We made changes accordingly (Line 27).
Round 2
Reviewer 1 Report
Thank you for the significant improvements made to the manuscript in this revision.
I have the following remaining comments regarding the response from the authors:
Comment 6: It is good that the authors considered other forms of keratitis in relation to miRNAs, and as such the above response should also be reflected very briefly at the start of each of the bacterial and viral keratitis subheadings in the article, for the readers to know that an attempt was made to search for, and review such literature
Also, the authors should revise the article to correct minor grammatical errors such as:
-Page 4, lines 171-172. It should read ‘Blood-feeding arthropods’ in place of blood feeing arthropods
-Page 4, line 180: can be detected by slit lamp examination, line 181: Larvae can migrate through the human body by
-Page 7, line 319: is predicted to have…
-Page 7, line 332. Should read occurs early in life…
-Page 9, line 443: Should read: that one miRNA can be involved in the pathogenesis…
Authors should provide a reference for the statement on page 7, lines 318-322. In addition, author should keep in mind that it is impossible to verify unpublished work, and as such, it is not very helpful to refer to such work when making a case.
On Page 10, lines 462-466, in regard to miRNAs in angiogenesis and inflammation, below is another up-to-date review article which should be referenced.
Mukwaya, Anthony, Lasse Jensen, Beatrice Peebo, and Neil Lagali. "MicroRNAs in the cornea: Role and implications for treatment of corneal neovascularization." The ocular surface (2019).
Author Response
Dear reviewers and editors:
Thank you again for your insightful review. We have made revisions accordingly. We “tracked changes” during the revision. So you can easily find the revisions in the text. Here is a summary of the changes made in response to your comments:
Reviewer 1:
Thank you for the significant improvements made to the manuscript in this revision.
I have the following remaining comments regarding the response from the authors:
Comment 6: It is good that the authors considered other forms of keratitis in relation to miRNAs, and as such the above response should also be reflected very briefly at the start of each of the bacterial and viral keratitis subheadings in the article, for the readers to know that an attempt was made to search for, and review such literature.
Answer (A): Thank you for the suggestion. Considering the flow of the manuscript, we have added at the end of miRNAs in bacterial keratitis on lines 330-335:
Staphylococcus aureus (S. aureus) is a leading cause of keratitis worldwide{Hazlett, 2016 #43733}(Hazlett, L.; Suvas, S.; McClellan, S.; Ekanayaka, S. Challenges of corneal infections. Expert Rev Ophthalmol 2016, 11, 285-297). However, there is no report on roles of miRNAs in S. aureus keratitis and keratitis caused by other bacteria (based on a Pubmed search with keywords of Staphylococcus aureus, keratitis, miRNA on 8/20/2019). This search result itself reflects an existing knowledge gap in miRNAs in bacterial keratitis.
We also added at the end of miRNAs in viral keratitis on lines 436-439:
A literature search on Pubmed (on 8/20/2019) by keywords of “viral, keratitis, miRNA” did not recover any more reports on miRNAs in viral keratitis other than HSK. This search result, again, reveals the shortage in our current knowledge on miRNAs in ocular infection and calls for more research effort in the future.
Also, the authors should revise the article to correct minor grammatical errors such as:
-Page 4, lines 171-172. It should read ‘Blood-feeding arthropods’ in place of blood feeing arthropods
A: change made accordingly. Thanks.
-Page 4, line 180: can be detected by slit lamp examination,
A: change made accordingly.
line 181: Larvae can migrate through the human body by
A: change made accordingly. But it doesn’t read well as: The larvae can migrate through human body by without provoking immune response. But if this reads well to you, that’s fine with me.
-Page 7, line 319: is predicted to have…
A: change made accordingly.
-Page 7, line 332. Should read occurs early in life…
A: change made accordingly by deleting “mostly”.
-Page 9, line 443: Should read: that one miRNA can be involved in the pathogenesis…
A: change made accordingly.
Authors should provide a reference for the statement on page 7, lines 318-322. In addition, author should keep in mind that it is impossible to verify unpublished work, and as such, it is not very helpful to refer to such work when making a case.
A: It seems that the editorial office may have made formatting changes. The line numbers appear to be different from what you referred to. I assume your request to “provide a reference for the statement on page 7, lines 318-322” is about “As a matter of fact, our preliminary trials of topical application of LNA-anti-miR-183/96/182 have showed promising results in controlling the severity of PA keratitis (Xu & Hazlett, unpublished data)” (now line 328-330 in the version I downloaded from publisher’s website). This is based on our unpublished work. Taking your advice that “author should keep in mind that it is impossible to verify unpublished work, and as such, it is not very helpful to refer to such work when making a case”, we have deleted this statement from the manuscript.
On Page 10, lines 462-466, in regard to miRNAs in angiogenesis and inflammation, below is another up-to-date review article which should be referenced. Mukwaya, Anthony, Lasse Jensen, Beatrice Peebo, and Neil Lagali. "MicroRNAs in the cornea: Role and implications for treatment of corneal neovascularization." The ocular surface (2019).
A: Added. However, because of the formatting by the editorial office, I couldn’t add this reference using the EndNote program I was using. I put the information on this reference, hoping that the editorial office will be able to add this reference properly in the text.
I hope you find our revised manuscript is acceptable for publication.
Best regards.
Shunbin Xu